# High Concordance between D:A:Dr and the Framingham Risk Score in Brazilians Living with HIV

**DOI:** 10.3390/v15020348

**Published:** 2023-01-26

**Authors:** Vitor Souza, Victória Valadares, Thais Dias, Carlos Brites

**Affiliations:** 1Department of Medicine, Medical School, Federal University of Bahia, Salvador 40110-060, BA, Brazil; 2Hospital Universitário Professor Edgard Santos, UFBA-EBSEHR, Salvador 40110-060, BA, Brazil

**Keywords:** cardiovascular risk, HIV, Brazil

## Abstract

People living with HIV (PLHIV) have twice the risk of developing cardiovascular diseases, making it essential to identify high cardiovascular risk (CVR). However, there is no validated CVR calculator for PLHIV in Brazil. We performed a cross-sectional study with 265 individuals living with HIV, aged 40 to 74 years, to assess the agreement between three CVR scores: Framingham Risk Score (FRS), Atherosclerotic Cardiovascular Disease (ASCVD) Risk Score, and a specific for PLHIV, Reduced Data Collection on Adverse Effects of Anti-HIV Drugs (D:A:Dr). We assessed agreement using the weighted Kappa coefficient and the Bland-Altman plot. The median age was 52 years (47–58), 58.9% were men, 34% were hypertensive and 8.3% had a detectable viral load. There was an almost perfect agreement between D:A:Dr x FRS (k = 0.82; 95% CI 0.77–0.87; *p* < 0.001), and substantial agreement between FRS vs. ASCVD (k = 0.74; 95% CI 0.69–0.79; *p* < 0.001) and between D:A:Dr vs. ASCVD (k = 0.70; 95% CI 0.64–0.76; *p* < 0.001). The Bland-Altman plot revealed greater discordance between scores as the CVR increased. Our results suggest that the FRS and the D:A:Dr are adequate to classify the CVR in this population, and the D:A:Dr score can be used as an alternative to the FRS in Brazil, as other international guidelines have already advocated.

## 1. Introduction

Morbidity and mortality resulting from HIV infection have been progressively reduced after the advent of combination antiretroviral therapy (cART) [1,2]. However, the longer survival provided by cART has been accompanied by a higher frequency of comorbidities, mainly associated with cardiovascular diseases (CVD), in a multifactorial manner that is still not well understood [3]. As a result, people living with HIV (PLHIV) have a relative risk of developing CVD about twice as high as that of the population without HIV [4].

The higher Cardiovascular Risk (CVR) of PLHIV is fueled by inflammation and chronic immune activation, in addition to the adverse effects of HIV treatment [5,6]. The use of some antiretroviral drugs, like those belonging to protease inhibitors (PI) and reverse transcriptase inhibitors classes, is related to lower levels of high-density lipoprotein (HDL), increase in total cholesterol (TC) and triglycerides (TG) and development of systemic arterial hypertension (AH). All these changes increase the CVR of these patients and the likelihood of outcomes such as acute myocardial infarction (AMI), stroke and congestive heart failure (CHF) [3,6,7]. 

Several instruments are used to assess CVR, allowing the implementation of primary prevention measures for these patients. Among these instruments are the Framingham Risk Score (FRS) and, more recently, the Atherosclerotic Cardiovascular Disease (ASCVD) risk score [8,9,10]. However, the early appearance of CVD in PLHIV led to the creation of a specific score for this population [11]. The Data-Collection on Adverse Effects of Anti-HIV Drugs (D:A:D) study proposed a score based on the analysis of 11 cohorts with PLHIV through the analysis of specific variables, including CD4+ cell count and exposure to certain antiretroviral therapies (ART). However, the difficulty in collecting data on the use of previous antiretrovirals led to the creation of a reduced version of D:A:D score (D:A:Dr), in which only the CD4+ cell count is taken into account [12]. 

Despite the existence of studies evaluating the accuracy of these scores, especially the FRS, it is not clear which one presents the best performance in PLHIV [13]. In general, the performance of calculators varies according to the characteristics of the population studied, which requires specific calibration [14]. In Brazil, the use of the FRS is recommended, but there is a lack of large studies analyzing its real performance in PLHIV [15]. In the present study, we evaluated the agreement between the most commonly applied CVR scores in PLHIV (D:A:Dr, ASCVD and global FRS), identifying the factors associated with the classification of high CVR and evaluating their potential impacts on the preventive use of statins, based on Brazilian standards and the high risk classification by these calculators.

## 2. Materials and Methods

### 2.1. Study Design and Population

This is a cross-sectional study comprising of 265 consecutive patients who attended the HIV/AIDS outpatient clinic, in previously scheduled appointments, at the Professor Edgard Santos University Hospital (HUPES) of the Federal University of Bahia (UFBA), between October 2020 and October 2021. The study is part of the Brazilian HIV-AIDS cohort (CoBRA), approved by the Research Ethics Committee of the Faculty of Medicine of Bahia (opinion number 1,035,826) and in accordance with the Declaration of Helsinki.

Socio-demographic variables were collected (age in years, sex, skin color and city of origin) and the participant’s medical history was reviewed to identify the occurrence of previous CVD (AMI, ischemic or hemorrhagic stroke, CHF, peripheral arterial disease and angina pectoris), diabetes mellitus (DM) (previous diagnosis and/or treatment), AH (previous diagnosis and/or medication use), use of lipid-lowering drugs (fibrate and/or statin), family history of early coronary artery disease (HECAD) (AMI or sudden death before age 55 in a male father or other first-degree relative, or before age 65 in a female mother or other first-degree relative); chronic kidney disease (previous diagnosis), smoking (current use, past use, if stopped for more than two last years, or never) and weekly alcohol use. Information on the time of diagnosis of HIV infection and the time of exposure to ART (in years), in addition to the mode of acquisition of HIV (heterosexual sexual relations, men who have sex with men (MSM) and others/not defined), were also recorded.

During the evaluation, weight (kg) and height (meters) were measured using an anthropometric scale. Systolic blood pressure (SBP) and diastolic blood pressure (DBP), in mmHg, were measured in the upper limbs (after five minutes of rest, in the supine position, with a digital sphygmomanometer). Body mass index (BMI) considered overweight and obesity as BMI ≥ 25 kg/m² and >30 kg/m², respectively. Abdominal circumference (AC) measurements were taken, considered to be increased if >102 cm in men and >88 cm in women. Lipid profile was considered altered if total cholesterol (TC) > 190 mg/dL, HDL < 40 mg/dL, LDL >130 mg/dL and triglycerides (TG) >150 mg/dL. The last CD4 cell count (cells/mL), the most recent HIV-RNA plasma viral load (values below 40 copies/mL were considered undetectable), in addition to fasting glucose (mg/dL) and serum creatinine (mg/dL) were recorded.

### 2.2. Risk Stratification

Three CVR scores were used, D:A:Dr, ASCVD and global FRS, all of which assessed the risk of cardiovascular events at 10 years. D:A:Dr was a score initially created to predict risk at 5 years, but an update to the assessment at 10 years was further provided, which was used [16]. Details on the characteristics of the scores used are shown in Table 1. 

To compare and assess the degree of agreement between the three scores, they were applied to all patients, with the CVR classification corresponding to the predetermined values for each score, as follows: Low CVR: D:A:Dr < 5%, ASCVD < 7.5%, FRS < 10%; Moderate CVR: D:A:Dr 5–10%, ASCVD 7.5–19.9%, FRS 10–20%; High CVR: D:A:Dr > 10%, ASCVD ≥ 20%, FRS >20%. 

### 2.3. Exclusion Criteria 

Patients not infected with HIV, not using ART, who attended the outpatient clinic without previous examinations, those under 40 years of age and over 74 years of age (common age limits to the scores used), non-Brazilians, or who had CVD were excluded from the study, as well as those patients who were unable to understand or to provide an informed consent form. 

### 2.4. Statistical Analysis

The sample size was calculated based on the 20% prevalence estimate of high CVR by the FRS [11], with a confidence level of 95%, an admissible error of 5% and a 80% power, resulting in a minimum sample of 246 participants. Statistical analyzes were performed using Statistical Package for Social Sciences (SPSS Inc., Chicago, IL, USA), version 25.0 for Windows. Categorical variables were expressed as absolute and relative frequencies, while normally distributed quantitative variables were described as mean and standard deviation and non-normal quantitative variables as median and interquartile range. The normality of the variables was tested using the Kolmogorov-Smirnov test. 

To compare the categorical variables of identification and clinical examination and the CVR stratification instruments, the Chi-Square test was used. For the evaluation between continuous variables of clinical examination and laboratory tests and the CVR stratification instruments, we used the Kruskal-Wallis test. We evaluated, in an univariate analysis, the association of specific variables with the stratification of RCV by calculators. 

### 2.5. Agreement between the CVR Stratification Instruments

For assessment of agreement between the CVR stratification instruments, a pairwise analysis was performed: ASCVD vs. FRS, D:A:Dr vs. ASCVD, and D:A:Dr vs. FRS. The level of agreement was assessed using the quadratic weighted Kappa test, as it takes into account the greater impact of belonging to the high-risk group. We considered weak agreement if Kappa values were between 0.21–0.40, moderate between 0.41–0.60, substantial between 0.61–0.80 and almost perfect agreement between 0.81–1.00, as proposed by Landis and Koch [17]. The difference in the 10-year predicted risk was also expressed through data dispersion in a Bland-Altman plot [18]. Statistical significance was assumed for a *p* value lower than 0.05 for all tests.

### 2.6. Recommendation for the Use of Statins According to CPTG

The evaluation of the indication of preventive therapy with statins was carried out according to the Clinical Protocol and Therapeutic Guidelines (CPTG) for PLHIV, published in 2018 by the Brazilian Ministry of Health [15]. The analysis was performed on all patients who were not yet using lipid-lowering drugs. All those with LDL > 130 mg/dL and high CVR by FRS were considered suitable for treatment. Additionally, the indication of therapy was evaluated if the CPTG hypothetically recommended the use of ASCVD and D:A:Dr as CVR stratification tools. 

## 3. Results

### 3.1. Patient Characteristics

A total of 339 HIV-infected patients were analyzed during the study. Of these, 49 (14.4%) were excluded because their ages were not compatible with all scores. Of the remaining 290 patients, 15 (5.17%) had previous cardiovascular events and were also excluded. In addition, 10 (3.6%) patients had incomplete laboratory data and were also excluded, resulting in a final sample of 265 patients (Figure 1).

Table 2 shows the sociodemographic, clinical and laboratory data of the patients. The population studied was predominantly male (58.9%), racially miscegenated (46.0%), with a median age of 52 (47–58) years. Only 11.3% of the sample reported smoking, with a median smoking history of 11 (4.72–25.7) packs/year, 52.5% were sedentary and 50.2% were overweight or obese. As for previous comorbidities, 34% have AH, and 9.1% reported DM. Family history of early coronary heart disease was reported by 15.8% of patients. In total, about 80.8% of the sample had some altered lipid fraction.

Median time since HIV diagnosis was 15.5 (7–22) years and the median time of antiretroviral therapy was 15 (7–21) years. Most (52.8%) patients reported being MSM, 26.8% declared themselves as heterosexual and in 20.4% route of infection was undefined. The CD4+ cells count was ≥ 500 cells/mm^3^ for 67.9% of patients, 27.5% had a CD4+ cells count between 200 and 499 and in only 4.5% the count was <200 cells/mm^3^. Viral load was detectable in only 8.3% patients, median of 2014.5 (159.5 – 17479) copies/mL. Table 2 summarizes the main characteristics of patients, according to the CVR scores. Regarding the use of ART, 5.3% used Atazanavir, 1.1% Darunavir, 7.2% Ritonavir, 9.4% Dolutegravir, 0.4% Raltegravir, 1.9% Abacavir, 12.5% Tenofovir, 17.0% Lamivudine, 1.5% Zidovudine, 3.8% Efavirenz and 2.3% Nevirapine. (Table 2). 

### 3.2. Risk Stratification 

It is possible to observe in Table 2 that age, AC measurement, smoking, AH, DM, TG >150 mg/dL, glucose ≥100 mg/dL, SBP, DBP, MAP, time since diagnosis and use of antiretroviral therapy were variables associated with a higher CVR by the three calculators. Gender was statistically significant for the FRS and ASCVD, with the high-risk group consisting mostly of males in the FRS (71.9%) and females in the ASCVD (57.9%).

Figure 2 shows the prevalence of CVR estimated by the three scores. It was observed that the FRS classified most of the sample as having a high CVR (21.5%), followed by D:A:Dr (20%) and ASCVD (7.2%). The prevalence of classification in low, moderate and high CVR was similar between FRS and D:A:Dr. The 2 × 2 cross-classifications between the calculators are shown in Table 3.

### 3.3. Agreement between the CVR Stratification Instruments

The agreement between the D:A:Dr and the FRS was almost perfect (k = 0.82; 95% CI 0.77–0.87; *p* < 0.001), and substantial between the FRS and ASCVD (k = 0.74; 95% CI 0.69–0.79; *p* < 0.001) and between D:A:Dr and ASCVD (k = 0.70; 95% CI 0.64 -0.76; *p* < 0.001). The median predicted CVR at 10 years was 10.0% (5.60–18.40) in the FRS, 6.20% (3.35–11.30) in the ASCVD and 5.23% (2.85 -8.74) in D:A:Dr (Table 4). There was greater disagreement between the calculators as the CVR increased (Figure 3). 

### 3.4. Recommendation for the Use of Statins According to CPTG

Of the 265 patients, 33 were using lipid-lowering drugs, leaving 232 patients for analysis. It was observed that, when applying the CPTG indications, 25 patients would be able to start preventive therapy with statins. If the CPTG recommended the use of ASCVD and D:A:Dr calculators, 8 and 23 patients, respectively, would be eligible. Most (80%) patients who were candidates for statin therapy using the FRS would also be eligible when evaluated by the D:A:Dr, but only 32% would be eligible if applying the ASCVD score.

## 4. Discussion

Our results demonstrate that the use of FRS or D:A:Dr for CVR stratification in a Brazilian HIV-AIDS cohort composed mainly of men would result in a similar CVR classification, while the use of ASCVD would underestimate the CVR for PLHIV. A high frequency of high CVR attributed by the calculators was also observed, which reflects a high prevalence of CVR factors found in the studied group. Other Brazilian studies that showed a lower prevalence of high CVR included populations with more controlled traditional risk factors [13,19]. 

The CVR scores are formulated based on large cohorts, which analyze the main variables related to predetermined outcomes. Factors associated with the study site, time/period of analysis and degree of sample heterogeneity directly influence these results. Therefore, it is difficult to observe absolute agreements between scores, which can lead to significant differences in the classification of patients and underestimation or overestimation of the real CVR [14]. PLHIV are at greater risk of developing cardiovascular diseases than the general population, which justifies the need to more accurately identify the CVR presented by this population [4]. In our study, 5.17% of the entire sample had a history of previous cardiovascular events, including five AMI, one ischemic stroke, two hemorrhagic stroke, four CHF, two angina pectoris and one peripheral arterial disease. 

Dyslipidemia, frequently found in PLHIV, is associated with HIV infection and the use of cART [20]. In the D:A:D study, 22.2% of the patients had high TC, 33.8% had an increase in TG and 25.7% had a decrease in HDL [16]. We found a higher rate of dyslipidemia than that observed in other Brazilian studies [13,19], especially increased LDL levels, the main target for the prevention and treatment of cardiovascular disease. Another factor that may have contributed to the high detected CVR is the overweight and obesity present in about 50% of the study population. In the literature, obesity is already considered a factor that may be closely linked to the use of ART and to dyslipidemia itself [21,22]. 

Other important CVR factors were also frequently detected in the study population, such as AH, DM, early HECAD and smoking. Smoking, in particular, is usually observed in a higher frequency in PLHIV than in the population without HIV. However, its frequency is significantly influenced by the study site [3,4,5,6,7,8,9,10,11,12,13,14,15,16,17,18,19,20,21,22]. In our population, 11.3% were current smokers and 23.8% had already used tobacco, results similar to those found in other Brazilian studies [13,19]. A study carried out with Taiwanese patients revealed that smoking cessation, in individuals aged between 55–59 years, would lead to a decrease of 33.5% and 20% in the CVR calculated by the FRS and ASCVD, respectively [23]. Thus, even with the significant drop in the number of smokers in recent years, medical intervention is still essential in providing guidance on smoking cessation [22,23,24]. 

In Brazil, the Ministry of Health recommends the stratification of the CVR by the FRS, both in the initial approach of the patient with HIV and when changing the therapeutic regimen [15]. Our findings suggest that the FRS provides a stratification similar to D:A:Dr with an almost perfect agreement (﻿𝑘=0.82; 95% CI 0.77–0.87; *p* < 0.001) [17]. A study carried out in the United Kingdom and Ireland showed a moderate agreement between these calculators (𝑘=0.41; 95% CI 0.37–0.45; *p* < 0.001), but the study’s population characteristics were quite different from ours. They also used a larger sample, making it difficult to make a more precise comparison with the present study [25]. A direct comparison of such evaluations with the available literature is difficult, because many articles use Cohen’s Kappa instead of the weighted Kappa, which, by definition, would be the most appropriate due to the ordinal character of the CVR classification. In addition, greater disagreement was noted in the comparison between all calculators as the CVR increased. This characteristic seems to be a constant among studies that applied the same methodology [25,26,27]. 

The ASCVD has been increasingly used as a stratification tool for CVR [28]. However, when applied to PLHIV, it underestimates the patient’s real risk, since it does not take into account the chronic and inflammatory nature of the infection, as well as the use of ART. Like the ASCVD, the FRS does not take these variables into account, but some studies demonstrate that the FRS can more accurately predict the risk of cardiovascular events in this population [28,29]. Furthermore, it is necessary to consider that the outcomes used by these equations are slightly different from each other and these, therefore, need to be considered when choosing a score. The ASCVD [30], for example, does not take into account the development of CHF and peripheral arterial disease, unlike the FRS [31], which may explain the lower classification of high risk by the ASCVD, as found in this study. D:A:Dr [16], despite not evaluating these last two outcomes mentioned, includes the need for invasive coronary procedures by patients. 

The use of scores to predict risk in PLHIV has been widely discussed [22]. Evidence is sought of which would be the most accurate and whether their results could be reflected in the therapeutic decision. In our study, 33 patients were already using lipid-lowering drugs and another 25 would be candidates to receive statins according to the CPTG guidelines [15], which recommends stratification of the CVR by the FRS. If the CPTG recommended the use of D:A:Dr, the indication would be very similar to that of the FRS. However, when using the ASCVD, only a small portion of the patients would be suitable for the therapy. Large studies have already demonstrated that the use of ASCVD in PLHIV could result in undertreatment with statins, with a reduction in the prescription of statins in more than 65% of patients who had evidence of carotid atherosclerotic plaque and who, therefore, could benefit from preventive therapy [30,31,32]. 

In addition, in a Brazilian study that evaluated patients with subclinical atherosclerosis, it was observed that the FRS stratified nine times more patients as having low CVR when compared with the D:A:Dr [11]. These findings have a great impact on clinical practice, since many patients would remain without adequate treatment, or would be undertreated with statins, if they had their CVR calculated only by traditional risk scores, such as ASCVD or FRS. Thus, despite the high concordance between the FRS and the D:A:Dr found in this study, health professionals should be careful when guiding the therapeutic approach based only on a single cardiovascular risk score.

Our study must be interpreted in the context of some limitations. Due to the cross-sectional nature of the research, it was not possible to analyze the calibration and discrimination of scores in relation to the outcomes evaluated by each equation. Second, laboratory data were not collected from the same location, so the interpretation of results may have included some laboratory variability. In addition, the sample size did not allow for more sophisticated, subgroup analyses. Thus, the results found here can only be applied to populations with similar CVR. However, this is a study with a sample of individuals under regular follow-up, in a reference center and stable antiretroviral treatment, allowing a standardized evaluation and without risk of selection bias. In addition, this is the first study in Brazil to assess these scores, using the D:A:Dr model to predict CVR in 10 years, and its results may provide definitions on the use of these instruments in the country.

## 5. Conclusions

The predicted 10-year CVR in PLHIV is significantly high in Brazil. We observed an excellent agreement between the FRS and D:A:Dr, suggesting that both scores are able to similarly classify the CVR in this population. However, there is a need for prospective studies to ensure the calibration, discrimination and validation of D:A:Dr in the Brazilian population. Our results suggest that the incorporation of D:A:Dr, together with the FRS, in the CPTG, could be implemented in Brazil, in accordance with international guidelines. 

## Figures and Tables

**Figure 1 viruses-15-00348-f001:**
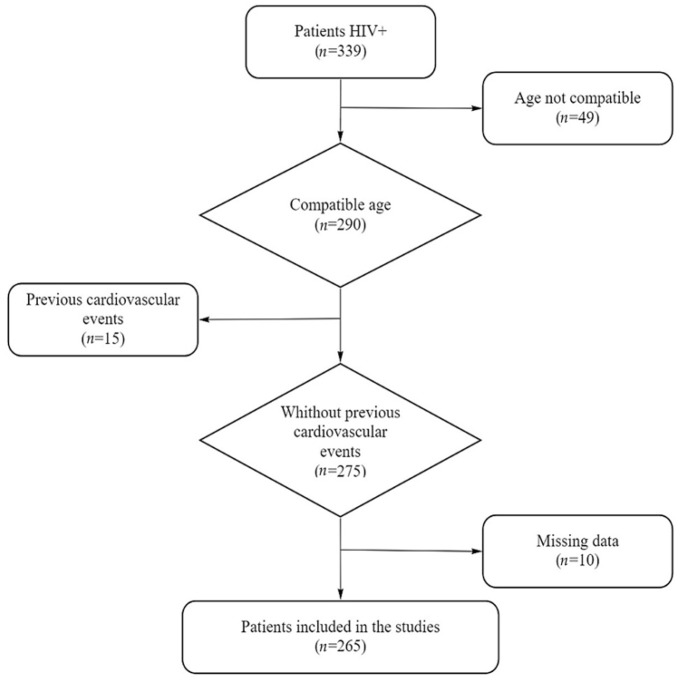
Study inclusion flowchart.

**Figure 2 viruses-15-00348-f002:**
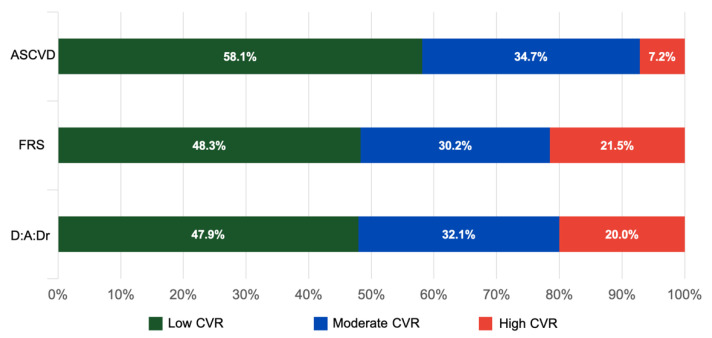
Prevalence of CVR according to ASCVD, FRS and D:A:Dr.

**Figure 3 viruses-15-00348-f003:**
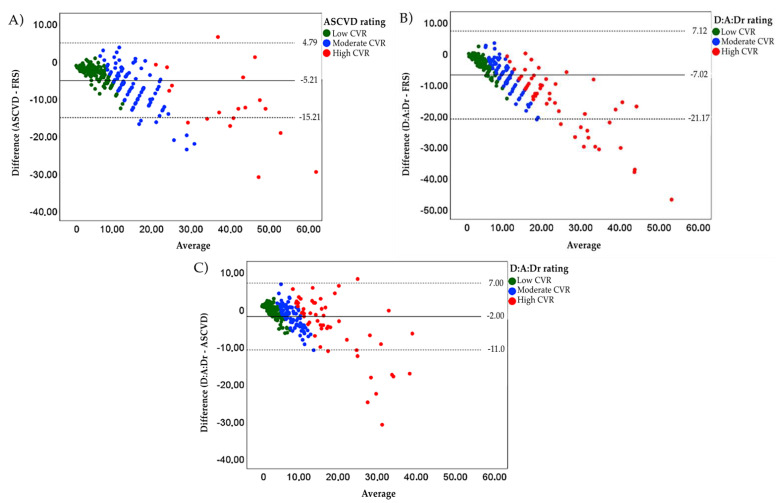
Bland-Altman plots comparing ASCVD, FRS and D:A:Dr. (**A**) The ASCVD risk score was on average 5.21% (±5.00%) lower than FRS; the limits of the difference were −15.21% and 4.79%. (**B**) The D:A:Dr risk score was on average 7.02% (±7.22%) lower than FRS; the limits of the difference were −21.17% and 7.12%. (**C**) The D:A:Dr risk score was on average 2.00% (±5.00%) lower than ASCVD; the limits of the difference were -11.00% and 7.00%.

**Table 1 viruses-15-00348-t001:** Characteristics of CVR stratification tools.

	FRS	ASCVD	D:A:Dr
**Cohort**	Framingham Heart Study	New pooled cohort	D:A:D study
**Predictors**	Age, systolic blood pressure, use of antihypertensive medication, current smoking, DM, HDL cholesterol.	Age, sex, race (white, African-American, others), systolic and diastolic BP, total cholesterol, HDL, DM, smoking, and antihypertensive therapy.	Gender, age, smoking, DM (diagnosed or on antidiabetic treatment), family history of early CVD, systolic BP, total cholesterol, HDL, CD4+ count.
**Age group**	30–75	40–74	18–75
**Cardiovascular outcomes**	Coronary heart disease, cerebrovascular and peripheral arterial disease, and heart failure.	First occurrence of nonfatal myocardial infarction, death due to coronary artery disease, and stroke.	Myocardial infarction, stroke, invasive coronary artery procedure or death due to coronary heart disease.

Legend: DM, diabetes mellitus; HDL, high density lipoprotein; CVD, cardiovascular disease; BP, blood pressure.

**Table 2 viruses-15-00348-t002:** Demographic, clinical and laboratory characteristics of the study population according to the CVR stratification instruments.

	n (%)	FRS			*p* *	ASCVD			*p* *	D:A:Dr			*p* *
		Low CVR	Moderate CVR	High CVR		Low CVR	Moderate CVR	High CVR		Low CVR	Moderate CVR	High CVR	
Sex (%)					**0.001**				**<0.001**				0.051
Male	156 (58.9)	60 (46.9)	55 (68.8)	41 (71.9)		79 (51.3)	69 (75.0)	8 (42.1)		65 (51.2)	56 (65.9)	35 (66.0)	
Female	109 (41.1)	68 (53.1)	25 (31.2)	16 (28.1)		75 (48.7)	23 (25.0)	11 (57.9)		62 (48.8)	29 (34.1)	18 (34.0)	
Race (%)					0.110				0.494				0.166
White	33 (12.5)	12 (9.4)	11 (13.8)	10 (17.5)		17 (11.0)	13 (14.1)	3 (15.8)		12 (9.4)	11 (12.9)	10 (18.9)	
Black	110 (41.5)	63 (49.2)	30 (35.7)	17 (29.8)		71 (46.1)	33 (35.9)	6 (31.6)		61 (48.0)	33 (38.8)	16 (30.2)	
Miscegenated	122 (46.0)	53 (41.4)	39 (48.8)	30 (52.6)		66 (42.9)	46 (50.0)	10 (52.6)		54 (42.5)	41 (48.2)	27 (50.9)	
Age †	52 (47–58)	48 (43.2–52.7)	55 (50–60.7)	60 (56–66.5)	**<0.001**	49 (44–53)	58 (53.2–65)	66 (58-69)	**<0.001**	47 (43–52)	56 (50–60)	64 (56.5–68)	**<0.001**
BMI (%)					0.581				0.272				0.907
Overweight	83 (31.3)	38 (29.7)	26 (32.5)	19 (33.3)		51 (33.1)	23 (25.0)	9 (47.4)		40 (31.5)	24 (28.2)	19 (35.8)	
Obesity	50 (18.9)	20 (15.6)	17 (21.3)	13 (22.8)		26 (16.9)	20 (21.7)	4 (21.1)		25 (19.7)	16 (18.8)	9 (17.0)	
Changed AC (%) (n = 259)	85 (32.8)	37 (29.6)	23 (29.9)	25 (43.9)	0.132	49 (32.5)	25 (28.1)	11 (57.9)	**0.042**	39 (31.7)	25 (29.8)	21 (40.4)	**0.412**
Smoking (%)					**<0.001**				**0.001**				**<0.001**
Never	172 (64.9)	96 (55.8)	51 (29.6)	25 (14.5)		116 (75.3)	47 (51.0)	9 (47.3)		105 (82.6)	46 (54.1)	21 (39.6)	
Past	63 (23.8)	19 (30.1)	26 (41.2)	18 (28.5)		25 (16.2)	31 (33.6)	7 (36.8)		17 (13.3)	25 (29.4)	21 (399.6)	
Current	30 (11.3)	13 (43.3)	3 (10)	14 (46.7)		13 (8.4)	14 (15.2)	3 (15.7)		5 (3.9)	14 (16.4)	11 (20.7)	
Alcohol abuse (%)	68 (25.7)	36 (52.9)	19 (27.9)	13 (19.1)	0.669	39 (57.3)	25 (36.7)	4 (5.8)	0.847	38 (55.8)	19 (27.9)	11 (16.1)	0.307
Sedentary life (%)	139 (52.5)	62 (44.6)	43 (30.9)	34 (24.4)	0.356	84 (60.4)	45 (32.3)	10 (7.1)	0.693	62 (44.6)	47 (33.8)	30 (21.5)	0.518
Previous comorbidities (%)													
AH	90 (34.0)	17 (18.8)	35 (38.8)	38 (42.2)	**<0.001**	23 (25.5)	50 (55.5)	17 (18.8)	**<0.001**	25 (27.7)	36 (40.0)	29 (32.2)	**<0.001**
DM	24 (9.1)	1 (4.1)	9 (37.5)	14 (56.0)	**<0.001**	2 (8.3)	12 (50.0)	10 (41.6)	**<0.001**	2 (8.3)	7 (29.1)	15 (62.5)	**<0.001**
CKD	6 (2.3)	2 (33.3)	2 (33.3)	2 (33.3)	0.703	2 (33.3)	3 (50.0)	1 (16.6)	0.400	1 (16.6)	3 (50.0)	2 (33.3)	0.299
Family history of early CAD (%)	42 (15.8)	22 (52.3)	12 (28.5)	8 (19.0)	0.837	27 (64.2)	13 (30.9)	2 (4.7)	0.627	15 (35.7)	18 (42.8)	9 (21.1)	0.182
Use of lipid-lowering drugs (%)	33 (12.5)	9 (27.2)	17 (51.5)	7 (21.2)	**0.010**	14 (42.2)	14 (42.2)	5 (15.1)	0.061	9 (27.2)	15 (45.4)	9 (27.2)	**0.040**
Changed lipid profile (%)													
CT > 190 mg/dL	149 (56.2)	64 (50.0)	47 (58.8)	38 (66.7)	0.093	84 (54.5)	49 (53.3)	16 (84.2)	0.038	60 (47.2)	51 (60.0)	38 (71.7)	**0.007**
HDL < 40 mg/dL	84 (31.7)	32 (25.0)	27 (33.8)	25 (43.9)	**0.035**	43 (27.9)	31 (33.7)	10 (52.6)	0.081	36 (28.3)	28 (32.9)	20 (37.7)	0.447
LDL > 130 mg/dL	110 (41.5)	48 (37.5)	34 (42.5)	28 (49.1)	0.326	64 (41.6)	35 (38.0)	11 (57.9)	0.279	45 (35.4)	39 (45.9)	26 (49.1)	0.146
TG > 150 mg/dL	118 (44.5)	41 (32.0)	41 (51.2)	36 (63.2)	**<0.001**	56 (36.4)	47 (51.1)	15 (78.9)	**0.001**	45 (35.4)	36 (42.4)	37 (69.8)	**<0.001**
Glucose ≥100 mg/dL (%) (n = 254)	94 (37.0)	34 (27.2)	35 (46.1)	25 (47.2)	**0.006**	43 (28.9)	40 (46.0)	11 (61.1)	**0.003**	35 (28.2)	35 (42.7)	24 (50.0)	**0.013**
Creatinine (mg/dL)† (n = 249)	0,92 (0.8–1.1)	0.9 (0.8–1.1)	1 (0.8–1.1)	1 (0.8–1.2)	**0.008**	0.9 (0.8–1.1)	1.0 (0.8–1.2)	1.1 (0.8–1.2)	**0.006**	1.0 (0.8–1.1)	0.9 (0.8–1.1)	1.0 (0.9–1.2)	0.097
Blood pressure													
SBP (mmHg) †	130 (120–140)	120 (110–130)	130 (120–140)	14 (130–160)	**<0.001**	120 (110–130)	130 (120–140)	160 (150–180)	**<0.001**	120 (110–130)	130 (120–140)	140 (129.5–160)	**<0.001**
DBP (mmHg) †	80 (70–81)	80 (70–80)	80 (70–90)	80 (80–90)	**<0.001**	80 (70–90)	80 (70–80)	90 (80–100)	**<0.001**	80 (70–80)	80 (70–90)	80 (70–90)	**<0.001**
MBP (mmHg) †	93.3(86.7–103.3)	90 (83.3–96.7)	96,7 (90–106.7)	103.3 (96.7–113.3)	**<0.001**	93.3 (83.3–96.7)	96.7 (90–106.7)	113.3 (103.3–126.7)	**<0.001**	93.3 (83.3–96.7)	96.7 (90-103)	103.3 (91.7–113.3)	**<0.001**
Time since diagnosis †	15.5 (7–22)	11 (5–21)	16.5 (11–22.7)	18 (11–22)	**0.003**	11.5 (6–21)	18 (11–22)	21 (13–27)	**<0.001**	12 (6–21)	16 (9.5–22)	19 (11–22)	**0.013**
Time of antiretroviral therapy †	15 (7–21)	10.5 (5–21)	15 (10.2–21)	17 (10–21.5)	**0.003**	11 (5–21)	17 (11–21)	20 (11–24)	**<0.001**	11 (6–21)	15 (9–21)	18 (10–21)	**0.015**
Sexual orientation (%)					0.543				0.345				0.852
Heterosexual	71 (26.8)	29 (10.9)	25 (9.4)	17 (6.4)		42 (15.8)	26 (9.8)	3 (1.1)		35 (13.2)	21 (7.9)	15 (5.7)	
MSM	140 (52.8)	73 (27.5)	37 (14.0)	30 (11.3)		83 (30.9)	44 (16.6)	14 (5.3)		65 (24.5)	45 (17.0)	30 (11.3)	
Not defined	54 (20.4)	26 (9.8)	18 (6.8)	10 (3.8)		30 (11.3)	22 (8.3)	2 (0.8)		27 (10.2)	19 (7.2)	8 (3.0)	
CD4+ (cells/mL) (%)					0.490				0.329				0.929
<200	12 (4.5)	7 (5.4)	2 (3)	3 (4.1)		7 (4.5)	5 (5.4)	0 (0.0)		5 (3.9)	4 (4.7)	3 (5.6)	
200–499	73 (27.5)	40 (31.2)	21 (32.3)	12 (16.6)		44 (28.5)	27 (29.3)	2 (10.5)		36 (28.3)	21 (24.7)	16 (30.1)	
﻿≥500	180 (67.9)	81 (63.2)	42 (64.6)	57 (79.1)		103 (66.8)	60 (65.2)	17 (89.4)		86 (67.7)	60 (70.5)	34 (64.1)	
Detectable viral load	22 (8.3)	12 (54.5)	9 (40.9)	1 (4.5)	0.089	15 (68.2)	7 (31.8)	0 (0.0)	0.493	13 (59.1)	7 (31.8)	2 (9.1)	0.413
Total	265	128 (48.3)	80 (30.2)	57 (21.5)		154 (58.1)	92 (34.7)	19 (7.2)		127 (47.9)	85 (32.1)	53 (20.0)	

Legend: * = χ2 test or Fisher’s exact test for categorical variables and Kruskal-Wallis for continuous variables. † = Median (Interquartile Range). BMI, body mass index; AC, abdominal circumference; AH, arterial hypertension; DM, diabetes mellitus; CKD, chronic kidney disease, CAD, coronary artery disease; CT, total cholesterol; HDL, high-density lipoprotein; LDL, low-density lipoprotein; TG, triglycerides; SBP, systolic blood pressure; DBP, diastolic blood pressure; MBP, mean arterial pressure; MSM, men who have sex with men.

**Table 3 viruses-15-00348-t003:** Cross-rating between risk ratings by calculators.

		FRS			ASCVD		
		Low CVR (%)	Moderate CVR (%)	High CVR (%)	Low CVR (%)	Moderate CVR (%)	High CVR (%)
**D:A:Dr**	Low CVR (%)	110 (41.5)	17 (6.4)	0 (0.0)	118 (44.5)	9 (3.4)	0 (0.0)
	Moderate CVR (%)	18 (6.8)	54 (20.4)	13 (4.9)	35 (13.2)	50 (18.9)	0 (0.0)
	High CVR (%)	0 (0.0)	9 (3.4)	44 (16.6)	1 (0.4)	33 (12.5)	19 (7.2)
**ASCVD**	Low CVR (%)	123 (46.4)	5 (1.9)	0 (0.0)	-	-	-
	Moderate CVR (%)	31 (11.7)	49 (18.5)	0 (0.0)	-	-	-
	High CVR (%)	0 (0.0)	38 (14.3)	19 (7.2)	-	-	-

Legend: CVR, cardiovascular risk.

**Table 4 viruses-15-00348-t004:** Agreement rate between the thee CVR stratification tools.

	CVR Score		
	FRS	ASCVD	D:A:Dr
Expected risk in 10 years (%) (median (IQR))	10.00 (5.60–18.40)	6.20 (3.35–11.30)	5.23 (2.85–8.74)
Agreement between scores			
**FRS**			
Observed agreement (%)	-	72.1	78.5
Weighted kappa (CI 95%)	-	0.74 (0.69–0.79)	0.82 (0.77–0.87)
*p* value	-	*p* < 0.001	*p* < 0.001
**ASCVD**			
Observed agreement (%)	-	-	70.6
Weighted kappa (CI 95%)	-	-	0.70 (0.64–0.76)
*p* value	-	-	*p* < 0.001

Legenda: IQR, interquartile range; CI, confidence interval; CVR; cardiovascular risk.

## Data Availability

Not applicable.

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
