# Peer review of "High Concordance between D:A:Dr and the Framingham Risk Score in Brazilians Living with HIV"

_viruses, 2023, doi:10.3390/v15020348_

Round 1
Reviewer 1 Report
Vitor Souza and colleagues present a paper comparing three instruments for calculating cardiovascular disease risk for people living with HIV (PLHIV) in Brazil. The instruments were Framingham Risk Score (FRS), Atherosclerotic Cardiovascular Disease (ASCVD) Risk Score, and the Reduced Data Collection on Adverse Effects of Anti-HIV Drugs (D:A:Dr). The population was 265 PLHIV on ART aged 40 to 74. Using the quadratic weighted Kappa test and Bland-Altman plots, the authors show concordance across all three instruments, although increased variability was observed in people with higher risk for CVD. The bottom line is that the D:A:Dr is a suitable alternative to the FRS while the ASCVD may underestimate the risk of developing CVD in PLHIV in Brazil. This is a very clear and cogent study however, it is not the first study to examine CVD risk estimates in PLHIV in Brazil. Despite these previous studies it is important to understand how these instruments work across different populations. There are several issues with the paper in its current form that should be addressed.
1. It is surprising that obesity, altered lipid profile, and time on ART were not associated with CVD in the multivariate analysis. Were they associated in a univariate analysis? This should be discussed.
2. This paper had a significantly lower number of individuals included in the analysis compared to other referenced studies in the UK (ref 25), Peru (ref 26), and Thailand (ref 27). What is the power to detect differences, and could this explain the lack of associations between obesity, dyslipidemia, and time on ART with CVR?
3. In table 5 there is no association between ASCVD and FRS – this should be discussed.
4. The authors mention “inflammation and chronic immune activation” on line 31 of the introduction. Have you measured any soluble factors associated with inflammation or immune activation in this cohort?
5. More detail is needed on the selection of the statistical test and why the weighed Kappa test was used and not Cohen’s Kappa test.
Minor issues:
· Grammar: Line 179 “Being male was the main associated with a high CVR in the FRS”
· Grammar: Line 61 “This is a cross-sectional, observational.”
· Wording Line 27: “and almost perfect agreement” and also on line 258, does not sound very scientific – maybe a high level of agreement?
Author Response
We thank reviewers for their comments and questions on our paper, that significantly improved the scientific quality of our manuscript. Please, see below a point-by-point answers to all reviewers’ questions/comments.
REVIEWER 01:
- It is surprising that obesity, altered lipid profile, and time on ART were not associated with CVD in the multivariate analysis. Were they associated in a univariate analysis? This should be discussed.
R: In the univariate analysis, these variables were associated with a high CVR classification. Considering the questions pointed out by reviewer 2 on the validity of multivariate analysis, we decided to exclude the multivariate analysis table (see answer 12 to reviewer 2).
- This paper had a significantly lower number of individuals included in the analysis compared to other referenced studies in the UK (ref 25), Peru (ref 26), and Thailand (ref 27). What is the power to detect differences, and could this explain the lack of associations between obesity, dyslipidemia, and time on ART with CVR?
R: Although our work has a lower number of individuals in comparison with the mentioned studies, we calculated the sample size necessary to identify differences in relation to the agreement between the scores, with a margin of error of 5% and estimation of the proportion of high risk in the population of 20%, resulting in a minimum N of 246 patients, considering a 80% power to detect differences. We agree that it lacks power to detect the potential association of obesity, dyslipidemia, and time on ART with the chance of belonging to the high CVR group in a multivariate analysis, but these outcomes were not the main objectives of the work.
- In table 4 there is no association between ASCVD and FRS – this should be discussed.
R: Table 4 shows that there is a significant association between ASCVD and FRS with a substantial concordance between both scores (k=0.74; 95% CI 0.69-0.79; p<0.001)
- The authors mention “inflammation and chronic immune activation” on line 31 of the introduction. Have you measured any soluble factors associated with inflammation or immune activation in this cohort?
R: Unfortunately, soluble factors associated with inflammation and immune activation were not evaluated.
- More detail is needed on the selection of the statistical test and why the weighed Kappa test was used and not Cohen’s Kappa test.
R: The information was added to methods.
- Grammar: Line 179 “Being male was the main associated with a high CVR in the FRS”
R: Fixed!
- Grammar: Line 61 “This is a cross-sectional, observational.”
R: Fixed!
- Wording Line 27: “and almost perfect agreement” and also on line 258, does not sound very scientific – maybe a high level of agreement?
R: The term “almost perfect” is the original term of the classification proposed by Landis and Koch for the interpretation of the Kappa value between 0.81-1.00. Therefore, we used the same term to describe classification.
Reviewer 2 Report
- P2-L68 “City of origin” is misspelled.
- In the exclusion criteria, I would specify if there were any issues with the nationality of the participants. In other words, the Authors should specify if in the study only Brazilians were included (or not).
- Based on the same principle according to which patients not on ART were excluded from the analysis, also viraemic persons should be excluded as too different in terms of immunoinflammatory compared to persons with an undetectable viral load.
- A clear definition of detectable viremia (or undetectable) should be provided in the main text or the table.
- Table 2: I would change the title to “Baseline clinical, demographic and laboratory characteristics….”
- Table 2: I would also add parameters regarding the HIV history such as: Median Zenith HIV RNA, Previous AIDS event, Current ART regimen, and Median years of ART.
- Table 2: The first decimal should always be reported in all the percentages (even if they are 0).
- Table 3-4 and Figure 2: Use point (.) instead of commas (,) to separate decimals.
- Authors should double-check the number of decimals after the point: it should be the same in all text
- Table 4: The coefficient B should be omitted because it is already reported as exponential (OR) in the next column
- Table 4: “Variaveis” it’s not written in English.
- Table 4: I don’t understand the role of the multivariate model. The model explores the association between outcomes (having a high CVR) and different exposures (variables included in the model). The first problem is that it is unclear by which score the patients were classified as having a high CVR. Subsequently, it is unclear why those variables were brought into the final model and why other variables (like years of undetectability) were not. This aspect should be better justified in the methods section. Furthermore, the various items are associated by definition with the score to which they belong. Therefore, I would not present the variables divided by the score. Finally, the model is underpowered with probable multicollinearity problems. The confidence intervals are too wide, and the resulting information is weak. In this way, I think the multivariate model adds nothing to the discussion. I would suggest removing it or seeking the advice of an expert statistician.
- Among the reasons that the authors find to justify the underestimation of ASCVD is the fact that this score does not consider the nature of the infection and the use of ART. This is true, however, not even the other two scores (D:A:D r) consider these aspects in their calculation. Perhaps a clarification on this aspect would be necessary.
- The authors should better explain why the UK study showed a lower agreement between DAD and FRS. Did they use different cut-offs? In which aspect the two populations were different and how this had an impact on the difference between the score calculations?
- Not clear why in P12 L295 the Authors stated, “patients would remain without adequate treatment…if they had their CVR calculated by…FRS”. But the Authors stated that FRS and D:A:D highly agreed. Could you explain that? This sentence may contradict the conclusions of the work
Author Response
We thank reviewers for their comments and questions on our paper, that significantly improved the scientific quality of our manuscript. Please, see below a point-by-point answers to all reviewers’ questions/comments.
REVIEWER 2:
- P2-L68 “City of origin” is misspelled.
R: Adjusted.
- In the exclusion criteria, I would specify if there were any issues with the nationality of the participants. In other words, the Authors should specify if in the study only Brazilians were included (or not).
R: There weren’t participants of other nationalities (this information was added to the method section).
- Based on the same principle according to which patients not on ART were excluded from the analysis, also viraemic persons should be excluded as too different in terms of immunoinflammatory compared to persons with an undetectable viral load.
R: The decision to exclude the patients that weren’t on ART and keep the patients that could present detectable viral load was based on the assumption of the influence of the chronic use of ART on the calculated RCV by D:A:Dr.
- A clear definition of detectable viremia (or undetectable) should be provided in the main text or the table
R: Viral load under 40 copies/mL were considered as undetectable (information added on the method section).
- Table 2: I would change the title to “Baseline clinical, demographic and laboratory characteristics….”
R: We changed the title accordingly.
- Table 2: I would also add parameters regarding the HIV history such as: Median Zenith HIV RNA, Previous AIDS event, Current ART regimen, and Median years of ART.
R: We added information on previous ART use, because we agree with their potential impact on CV outcomes. However, we didn’t collect data about the previous AIDS events, as they probably have little impact on the current situation of patients on stable ART, most of them presenting with a normal CD4 count.
- Table 2: The first decimal should always be reported in all the percentages (even if they are 0).
R: The decimals were adjusted in all the text and tables.
- Table 3-4 and Figure 2: Use point (.) instead of commas (,) to separate decimals.
R: The commas were changed for the points.
- Authors should double-check the number of decimals after the point: it should be the same in all text
R: All the numbers were checked in the tables and text.
- Table 4: The coefficient B should be omitted because it is already reported as exponential (OR) in the next column
R: The table with multivariate analysis was removed from the article (see answer 12).
- Table 4: “Variaveis” it’s not written in English.
R: The word was translated.
- Table 4: I don’t understand the role of the multivariate model. The model explores the association between outcomes (having a high CVR) and different exposures (variables included in the model). The first problem is that it is unclear by which score the patients were classified as having a high CVR. Subsequently, it is unclear why those variables were brought into the final model and why other variables (like years of undetectability) were not. This aspect should be better justified in the methods section. Furthermore, the various items are associated by definition with the score to which they belong. Therefore, I would not present the variables divided by the score. Finally, the model is underpowered with probable multicollinearity problems. The confidence intervals are too wide, and the resulting information is weak. In this way, I think the multivariate model adds nothing to the discussion. I would suggest removing it or seeking the advice of an expert statistician.
R: We agree with the reviewer, and decided to suppress the table with multivariable analysis.
- Among the reasons that the authors find to justify the underestimation of ASCVD is the fact that this score does not consider the nature of the infection and the use of ART. This is true, however, not even the other two scores (D:A:Dr) consider these aspects in their calculation. Perhaps a clarification on this aspect would be necessary.
R: The D:A:Dr was created on a specific cohort of HIV living patients, and is the only score that considers CD4 count as a predictor variable. Regarding the ASCVD, it does not take into account outcomes assessed by the FRS, which could underestimate CVR in comparison to the other scores.
- The authors should better explain why the UK study showed a lower agreement between DAD and FRS. Did they use different cut-offs? In which aspect the two populations were different and how this had an impact on the difference between the score calculations?
R: The explanation was added to the paragraph.
- Not clear why in P12 L295 the Authors stated, “patients would remain without adequate treatment…if they had their CVR calculated by…FRS”. But the Authors stated that FRS and D:A:D highly agreed. Could you explain that? This sentence may contradict the conclusions of the work
R: Indeed, the way the sentence was written could lead to that interpretation. We just wanted to highlight that a perfect risk score for stratifying these patients is not available, and caution is necessary to guide the clinical judgment and the therapeutic approach based on the use of a single risk score.
Reviewer 3 Report
Dear authors.
After a careful examination of your work, I can say that I found it very interesting and a valuable addition to the current literature on this topic.
In the introduction, you state that the Framingham score is proposed to evaluate the cardiovascular risk in PLWH by your local guidelines, however, in the current literature there are findings saying traditional scores (Framingham included) may underestimate the CV risk in this kind of population (Schulz CA et al. Prediction of future cardiovascular events by Framingham, SCORE and asCVD risk scores is less accurate in HIV-positive individuals from the HIV-HEART Study compared with the general population. PMID: 34028959.), therefore I think is more appropriate to specify it in the introduction and is important to say that the concordance with the D:A:D score can hardly support the use of the latter to assess the CV risk in PLWH.
Besides that, I believe that your work may be eligible for publication in this journal after some minor adjustments:
1) in line 147 the "black" percentage is missing, it seems that the "miscegenated" percentage is referring to both (moreover miscegenated is spelled wrong).
2) The results do not mention "blood pressure" findings.
3) The "time since diagnosis" IQ range is different between table 2 and the results.
4) Percentage of "not defined" "sexual orientation" in table 2 is miscalculated.
5) Some tables are mentioned in the results section (tables 8 and 9), but I cannot find them anywhere in the paper.
Author Response
We thank reviewers for their comments and questions on our paper, that significantly improved the scientific quality of our manuscript. Please, see below a point-by-point answers to all reviewers’ questions/comments.
REVIEWER 3
- In line 147 the "black" percentage is missing, it seems that the "miscegenated" percentage is referring to both (moreover miscegenated is spelled wrong).
R: To make it clearer we used only the percentage of the most prevalent ethnicity, the miscegenated.
- The results do not mention "blood pressure" findings
R: The findings on “blood pressure” are shown in table 2.
- The "time since diagnosis" IQ range is different between table 2 and the results.
R: We corrected the values.
- Percentage of "not defined" "sexual orientation" in table 2 is miscalculated.
R: Right, there was a typo in the percentage of “not defined”, it was corrected.
- Some tables are mentioned in the results section (tables 8 and 9), but I cannot find them anywhere in the paper.
R: Tables 8 and 9 corresponded to tables not included on the final version of this paper, we deleted those mentions from text.
Round 2
Reviewer 2 Report
ok